# YouTube as a source of information on space maintainers for parents and patients

**Belen Şirinoğlu Çapan** [ID]*

Department of Pediatric Dentistry, Faculty of Dentistry, Biruni University, Istanbul, Turkey

* bcapan@biruni.edu.tr

## Abstract

### Objective

In recent years the social media offers a handy platform for patients who want to receive medical information. The aim of this study is to assess the content of YouTube as an information source on space maintainers and to evaluate the efficiency of videos by parents and patients.

### Methods and findings

YouTube was searched for information using two keywords 'space maintainer' and 'space maintainers in pediatric dentistry'. Two dentists reviewed the first 200 videos for each search term. After exclusions, 52 videos were included for analysis. Demographics of videos, including the type of source, date of upload, length, viewers' interaction and viewing rate were evaluated. The overall usefulness of videos was scored according to the American Academy of Pediatric Dentistry Guideline based on 9-point scales. The mean usefulness score of evaluated videos was 4,4 ± 1,5 (range 1–8). There was a significant correlation between usefulness and video length (p<0,05). But there was no significant correlation between usefulness and other demographics. Most videos were uploaded by healthcare professionals (51.9%). Videos uploaded by individual users were less useful compared with videos uploaded by healthcare professionals or organizations. Most of the videos (88,5%) used representative images for visualizing.

### Conclusions

YouTube videos about space maintainers are useful for parents only to understand and visualize their function. However, it does not provide sufficient information about adverse effects and types of them. Dentists should warn their patients that videos on the internet may contain incorrect and limited information.

## Introduction

Dental caries is a very common chronic disease of childhood in the most developed and developing countries, including around 60–90% of school-aged children affected by dental caries

**Citation:** Çapan BŞ (2021) YouTube as a source of information on space maintainers for parents and patients. PLoS ONE 16(2): e0246431. https://doi.org/10.1371/journal.pone.0246431

**Data Availability Statement:** All relevant data are within the paper and its Supporting Information files.

**Funding:** The author received no specific funding for this work.

**Competing interests:** The authors have declared that no competing interests exist.

[1,2]. Although the incidence of dental caries decreases day by day as a result of the many preventive treatments, dental caries and tooth loss due to caries are still an important health problem especially in developing countries [3].

Space management after tooth loss plays an important role in dental practice. In 1887, Davenport [4] described the space loss resulting from the early loss of primary teeth. Tooth decay, trauma and congenital deficiency are among the common causes of tooth loss [5].

Space maintainers are fixed or removable appliances used to preserve arch length following the premature loss or elective extraction of a tooth. Dentists should give importance to space management in their patients, as loss of arc length can lead to problems such as crowding, ectopic eruption, impaction and cross-bite in teeth [6]. The use of space maintainers will prevent patients from further exposure to complex orthodontic treatments or tooth extractions. The age, oral hygiene and the cooperation of the patient have importance on choosing the type of space maintainer. Although, it is well known that the maintenance of these spaces prevents later complications such as crowding and malocclusion, the use of space maintainers has some adverse effects as an increased plaque accumulation, which can lead to dental caries and periodontal disease [3,6]. Therefore children using the appliance should be checked regularly and given oral hygiene training.

In order to prevent loss of arch length after early loss of primary teeth, dentists inform patients about space maintainers. However, nowadays, with the widespread use of the internet, the information of the dentists is not enough for the patients to accept the treatment. Patients often receive advice from friends and relatives about treatments and most often consult the internet. They read articles or watch videos on the internet about their health problems. The social media offers a handy platform for patients who want to receive medical information. In fact, more than 80% of internet browsing activities are for medical support and information [7,8]. The most commonly used video-sharing site YouTube accrues more than 2 billion views per day. On YouTube, every minute a new video is being uploaded and an average user spends at least 15 minutes per day [9,10].

Video-sharing platforms like YouTube are attractive to share medical or dental information as an interface where information can be stored and people have a quick and costless access to this information. However, not every video on YouTube contains accurate and sufficient information about a particular topic, and some may even contain incorrect information. Therefore, healthcare professionals should listen to the patients carefully and give them the correct information by emphasizing them again.

So far, several studies in the literature have evaluated YouTube content in various medical aspects including oral health [7–9,11,12]. To our knowledge, no studies have assessed YouTube content regarding space maintainers by determining how coherent the presented information is with current policies and guidelines. Therefore, the present study aimed at analyzing the content of YouTube videos regarding the etiology, health effects, treatment considerations and types of space maintainers and to evaluate the efficiency of videos by parents and patients.

## Materials and methods

This study was exempted from ethical approval due to its observational nature and the use of publicly accessible data.

### YouTube search

We searched YouTube (www.youtube.com) in October 2019 for videos related to space maintainers using the default settings; two discrete searches were performed: October 7, 2019, using the search term 'space maintainer' and October 14, 2019, using the search term 'space

maintainers in pediatric dentistry'. Previous researches in the literature reported that 95% of users conducting an online search on YouTube will watch no more than the first 60 videos of the output and would not continue to search after the first 5 pages. Most studies utilizing You-Tube as a search engine have used 60–200 videos [8,11,13]. In accordance with this data we viewed and analyzed the first 200 videos for each search term (i.e., a total of 400 videos). Links of videos were saved for future analyses.

## Selection and analyzes of videos

Two dentists (B.S.C. and P.S.) initially viewed the videos together to exclude irrelevant videos such as videos demonstrating technical procedures involved in space maintainer construction, non-English language videos, videos with no sound or headings, duplicate videos, advertisements, songs, and conference/dental school lectures. The remaining videos were analyzed independently by the same dentists to determine video demographics including video length, date, and source of upload, country of origin and numbers of total views, likes, and dislikes. The sources of upload were categorized as universities or professional organizations, health-care professionals, health information web sites, commercial (dental manufacturing companies) and other individual users.

Interaction index and viewing rate were calculated based on the methods described in the previous studies as the following [7,11]:

Interaction index,

$$\frac{number\ of\ likes - number\ of\ dislikes}{total\ number\ of\ views} x100\%$$

Viewing rate,

$$\frac{number\ views}{number\ of\ days\ since\ upload} x100\%$$

We assessed videos for the presence of content about etiology, general dental health effects, purpose, adverse health effects, treatment considerations, treatment modalities, the importance of follow-up appointments, the period of appliance use and the presence of representative images. Each video was given a score from 0 to 9 to indicate its usefulness in providing parents with adequate information (Table 1). A 'usefulness score' was devised to categorize videos as not useful, slightly useful, moderately useful, and very useful. Each aspect was given a score of 1 or 0 being consistent with the guidelines of the American Academy Pediatric Dentistry (AAPD) [14] or not. A score of 9 indicated that the video mentioned etiology, health effects, considerations and types of space maintainers and used representative clinical images to describe them. Score 0 indicated that the video does not contain any utilizable information about space management. Any disagreements between researchers in usefulness scoring were solved by reviewing the literature and determining how consistent the provided information is with the current AAPD guidelines [14].

## Statistical analysis

The data obtained in this study were analyzed with SPSS 17 program. Kolmogorov-Smirnov and Shapiro Wilk's tests were used to investigate the normal distribution of variables. When the differences between the groups were examined, Kruskal-Wallis and Mann Whitney U tests, which are nonparametric methods, were used for comparisons between the groups when the variables did not comply with the normal distribution. Multiple Regression analysis was

**Table 1. Usefulness score.**

| Usefullness Score | |
|---|---|
| **Scoring İtem** | **Score** |
| Etiology (early tooth loss, trauma etc.) | 1 |
| General dental health effects (malocclusion, crowding, overjet etc.) | 1 |
| Adverse health effects (caries, plaque accumulation etc.) | 1 |
| Purpose of space management | 1 |
| Tratment considerations | 1 |
| Treatment modalities (fixed, removable) | 1 |
| Importance of follow-up appointments | 1 |
| Period of aplliance use | 1 |
| The presence of representative images | 1 |
| **Total** | **9** |

Score 0 = not useful; scores 1–3 = slightly useful; scores 4–6 = moderately useful; scores 7–9 = very useful.

performed to measure how much the independent variable explained the dependent variable. With regression analysis, it was indicated which of the variables had an effect. Statistical significance was set p<0.05.

## Results

### The output of the YouTube search

The search term 'space maintainer' resulted in a total of 13360 videos and the search term 'space maintainers in pediatric dentistry' resulted in a total of 1445 videos. We evaluated the first 200 videos of the output for each search term. Of the 400 videos initially viewed, 348 were excluded (Fig 1).

### Characteristics of videos

The mean length of videos was 3,1 min ± 2,8 (range from 30 sec. to 12,2 min). The total number of views of space maintainer related videos was 719 967 and the mean number of views was 13 867 (range from 7 to 561 039). The overall mean number of likes was 67 (range from 0 to 2100), whilst the mean number of dislikes was 2 (range from 0 to 69). The mean viewing rate 10,9 views/day ± 51,4 (range: 0,017–365,97 views/day).

Most videos were uploaded by healthcare professionals (51.9%; n = 27), followed by universities/professional organizations (36,6%; n = 19), other individual users (5,8%; n = 3), health information Web sites (3,8%; n = 2) and commercial companies (1,9%; n = 1), respectively (Table 2). Most of the videos (n = 12; 24%) were uploaded in 2018.

The videos have been uploaded by users from a wide range of countries; including America, India, Europe, England, Canada, Philippines, Egypt, Nepal, Iran, and Saudi Arabia. The majority of videos were uploaded by users in the USA (38,6%; n = 20); fourteen videos were uploaded by India (26,9%) and seven by users in Europe (13,5%). The source of the upload of one video could not be determined.

### Content of videos

YouTube videos contained various information about etiology, purpose, health effects and treatment process of space maintainers. The majority of the videos described the etiology, purpose and general dental health effects in detail. Only four (7,7%) of them did not mention the

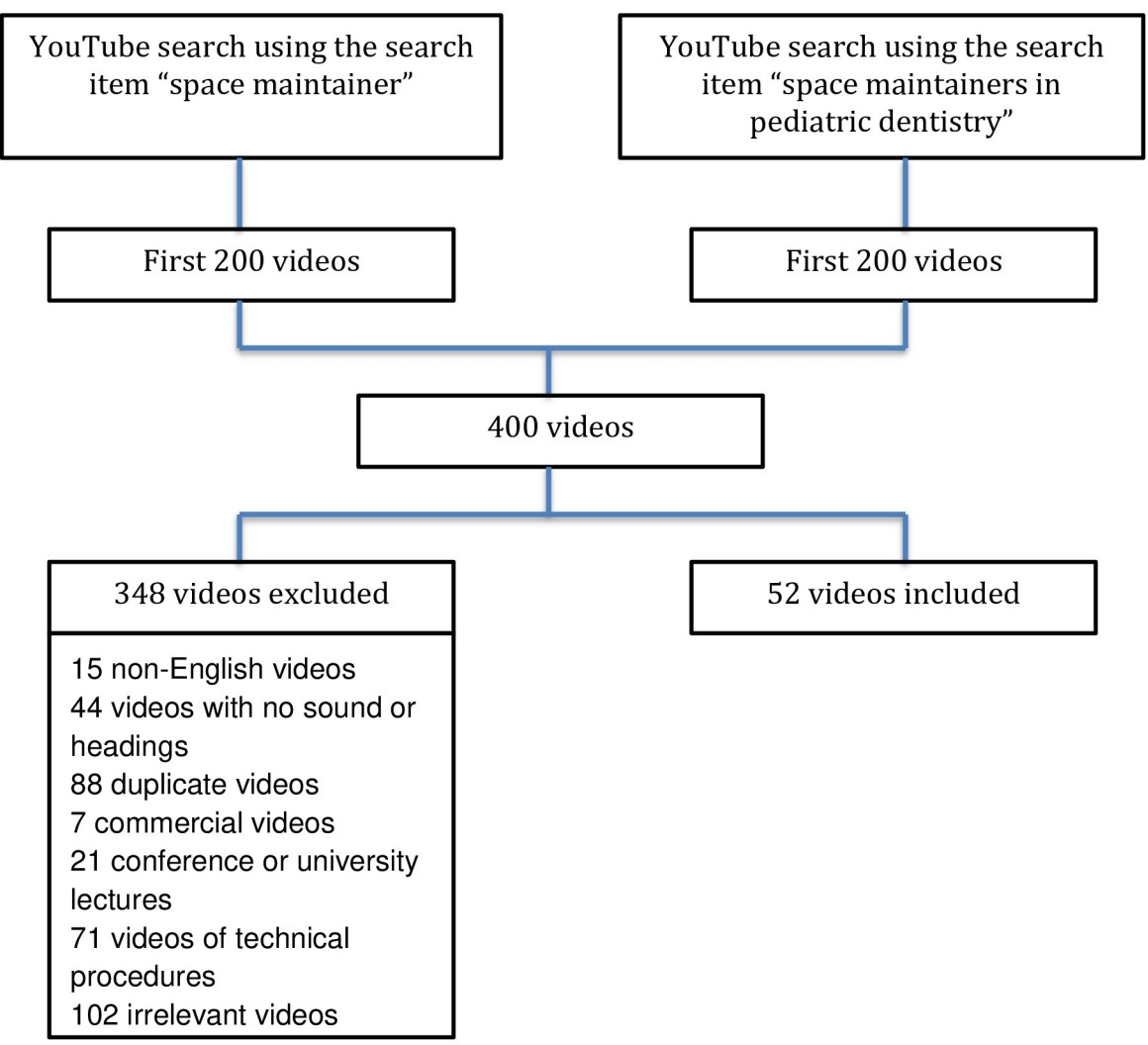

**Fig 1. YouTube video selection for analysis.**

purpose of the space maintainers and seven (13,5%) of them did not mention the etiology. Compared to purpose and etiology, there are more videos which was not describe the general dental health effects (23,1%; n = 12).

Most videos (88,5%; n = 46) contained representative images for the apparence of space maintainers. But only %30,8 (n = 16) of the videos shared different types of space maintainers visually and explained their indications. The rest explained the etiology, purpose and etc. with the appearance of only one single type space maintainer.

**Table 2. Upload source of YouTube videos on space maintainers.**

|  |  | n | % |
|---|---|---|---|
| **Upload Source** | Universities or professional organizations | 19 | 36,6 |
|  | Healthcare professionals | 27 | 51,9 |
|  | Health information Web sites | 2 | 3,8 |
|  | Other individual users | 3 | 5,8 |
|  | Commercial companies | 1 | 1,9 |

Table 3. Upload source according to the usefulness of content.

| Upload source | Usefulness n (%) | | | | |
|---|---|---|---|---|---|
| | Very Useful | Modaretely Useful | Slightly Useful | Not Useful | TOTAL |
| Universities or professional organizations | 1 (1,9%) | 14 (26,9%) | 4 (7,7%) | 0 | 19 (36,5%) |
| Healthcare professionals | 3 (5,8%) | 17 (32,7%) | 7 (13,5%) | 0 | 27 (51,9%) |
| Health information Web sites | 1 (1,9%) | 1 (1,9%) | 0 | 0 | 2 (3,8%) |
| Other individual users | 0 | 1 (1,9%) | 2 (3,8%) | 0 | 3 (5,8%) |
| Commercial companies | 0 | 1 (1,9%) | 0 | 0 | 1 (1,9%) |
| TOTAL | 5 (9,6%) | 34 (65,4%) | 13 (25%) | 0 | 52 (100%) |

Unfortunately, 11,5% (n = 6) of the videos express adverse effects. Only half of these explain all the adverse effects, the other half just stood on plaque accumulation and function loss of the appliance. Eight videos (15,4%) define treatment considerations. Half of these (50%; n = 4) were uploaded by healthcare professionals.

Only a few of the videos (%7,7; n = 4) describe the importance of follow-up controls. 34,6% (n = 18) of videos explained how long the space maintainers should be used.

## Usefulness and viewers' interaction

The mean interaction index score was 1,02% ±1,79 (range from 0 to 10,89%). A 9-point useful-ness score was devised to evaluate the usefulness of videos for the presence of content about etiology, purpose, adverse health effects, treatment considerations/planning and importance of follow-up appointments. The mean usefulness score of videos was 4,4 ±1,5 (range 1–8). There were 13 (25%) slightly useful videos, 34 (65,4%) moderately useful videos and 5 (9,6%) very useful videos. There were no zero (not useful) rated videos and 9-rated videos (a video, which got full score) (Table 3). All very useful videos except one, were uploaded by healthcare professionals or professional organizations. That one exceptional video was uploaded by a health information website that shares contents similar to shortened lecture notes, mostly to support the education of dentistry students.

Usefulness score of videos showed no significant correlation neither with viewing rate (r = 0,066, p>0,05), nor with interaction index (r = 0,009, p>0,05) score. However usefulness score of videos showed positive correlation with video length (r = 0,276; p<0,05) (Table 4).

No correlation found between upload source and views (p = 0.234), viewing rate (p = 0.619), interaction index score (p = 0,066) and usefulness score (p = 0,642). Although there was no statistically significant difference between upload source and usefulness score, the average usefulness score of videos uploaded by individual users (2.7) was significantly lower than other sources (Healthcare professionals 4.5; Universities 4.5; Health information Web sites 5.5; commercial company 5). The kappa value for the interobserver agreement was found 0.857.

Table 4. Comparison between characteristics and usefulness of YouTube videos on space maintainers.

| | | Usefullness Score |
|---|---|---|
| Video lenght | r | ,276* |
| | p | ,048 |
| Viewing Rate | r | ,066 |
| | p | ,644 |
| Interaction Index Score | r | ,009 |
| | p | ,952 |

## Discussion

Preventive dentistry is a control and treatment process that begins with the eruption of the first tooth in the mouth and continues until the end of life. The primary objective of preventive dentistry is to detect dental problems early and to take precautions. The preventive orthodontic/pedodontic approach includes space maintainers, grinding of primary tooth and plaque applications in crossbite cases and oral bad habits [15]. Space maintainence in children has often been accomplished inappropriately due to poor appliance design and poor case selection [16]. If space maintainers are placed at the right time with the right indication, they are highly effective in preventing problems caused by loss of arch length, such as malocclusion and crowding. In addition, the use of space maintainers obviates the need for later complex orthodontic treatment in terms of time and cost.

The importance of primary teeth should be understood by parents. After extraction of primary teeth due to caries and trauma, dentists inform parents about space maintainer application. However, due to both financial problems and insufficient information, parents do not understand the importance of treatment and the purpose of the space maintainer. Nowadays, parents use the internet to learn more about the topic they are curious about. Websites like YouTube, are being increasingly used by parents/patients as a source of health-related information and YouTube provides a convenient and free way to easily reach a large number of videos about dental treatments [13].

Previous studies in the literature have evaluated YouTube as a source of information about various dental/oral-health-related issues, such as root canal treatment, early childhood caries, mouth cancers and orthodontics [7,8,11,17]. The present study is the first to investigate the content of YouTube videos on space maintainers and to evaluate the efficiency of videos by parents and patients.

Video search in this study showed, that YouTube offers various information about space maintainers that ranges from highly specialized lectures to children's personal experiences with the treatment. According to the results, no statistically significant difference was found between usefulness score and upload source. Although there was no statistically significant difference, the scores of individual user videos were very low. Because these videos were mostly uploaded by children between the ages of 8–12, with limited information. Children were showing their space maintainers in these videos.

Etiology, purpose and general dental health effects of space maintainers were the most commonly described subjects in YouTube videos. For this reason, we think that YouTube videos about space maintainers can be enough to raise awareness of parents. According to the present study, very few of the videos mentioned the importance of follow-up controls and the adverse effects of space maintainers. However, the risk of newly developing caries and new tooth loss will increase if the control is not performed at the correct intervals. From this perspective, YouTube videos have limited usefulness.

The majority of the videos evaluated in this study included representative clinical images. Using representative clinical images of space maintainers is a applicable educational tool, which could help parents/patients to recognize the importance of arch length protection. Without representative images, it is difficult for patients to understand the definition of space maintainer and visualize it.

Similar to other popular social media platforms, YouTube allows every registered user to upload and share health-related videos free and without a referee check. The information source contained in these videos is often uncertain. This feature makes YouTube susceptible to publishing false and probably dangerous videos that are not based on scientific evidence [11]. Nevertheless, we found only a few misleading videos on space maintainers. On the contrary,

there were too many technical videos about space maintainer, which explain the construction of the appliance in detail. Both these detailed videos and misleading videos are confusing for parents/patients and do not provide useful information for them.

In the present study, we found no correlation between upload source and views, as well as video length, viewing rate, and interaction rate. This result is similar to studies in the literature [7,11,18].

Previous studies evaluating the contents of health-related videos on YouTube reported different results regarding viewers' interaction with videos [11,18–20]. According to the study results, there was no significant correlation between the videos' usefulness and the viewing rate or the interaction index score. This reveals another limitation for using YouTube as an information source about space maintainers because YouTube sorts its videos by relevance when used in default settings. The 'relevance' filter will show the most viewed and highly rated video at the top [8]. This means that if useful videos do not get enough rates, they won't be in the top of the list and might not be watched by viewers. We found a positive correlation between usefulness score and video length. That means the longer the duration of the video is, the more information can be explained. As a result, the video is getting more useful. When searching on YouTube, it also allows you to sort videos by criteria such as view count, upload date, rating and duration of the video in addition to relevance. Considering the results of our study, if the parents/patients sort the videos by duration, they will have the opportunity to watch more useful videos.

In the present study, videos were evaluated for the usefulness of information on space maintainers using a guideline-based scoring system that showed a satisfactory interobserver agreement. Usefulness scores range from 1–8 with a mean of 4,4 ±1,5, indicating not very low scores. Contrariwise to other studies evaluating YouTube videos regarding other health-related topics where the content was found to be generally poor and incorrect [7,17], our study shows that the content of YouTube videos was sufficient for basic information about space maintainers. According to our results, there wasn't any video that has a 9-point usefulness score. Actually, there were some videos that could get 9-point, but these videos were excluded because they were long lecture videos. People who search on YouTube can get more details by watching these videos, but it will be more difficult for patients to understand because the language they use contains more technical words. When we examined the excluded videos, the number of videos excluded in the second half was more than the number of videos that were excluded from the first half. This result may indicate that the number of duplicate and irrelevant videos increases as the number of pages increases.

Increasingly, patients are using the internet to get information on their disease and treatment modalities along with consulting reviews of physicians or dentists. For this purpose, they can access many contents such as videos about the experiences of patients, educational and / or advertising posts on the internet. That may seem to be harmless or even useful. The problem here is that source of the content on social media platforms (YouTube, Facebook, Instagram etc.) is not inspected, most of them are not refereed or evidence-based [21,22]. Therefore, the content is not reliable. In the study of Lavorgna et al. (21), almost all physicians have encountered patients who had already made a self-diagnosis on the Internet before coming to see the doctors. The authors sets forth that physicians should warn their patients against incorrect/ unreliable information on the internet and recommend reliable online sources. Studies from the USA, England and Italy in the literature report that more than half of dentists use social media, but they use social media mostly for advertising/marketing rather than communicating with or informing their patients [21,23,24]. Arnett et al. [25] showed that only 4.2% of the dental educators uploaded an informative video to YouTube.

Although social media is used by professionals mostly for marketing purposes, it is necessary to draw attention to giving correct instruction to patients/public within the framework of ethical rules of medicine and education. Of course, dentists can do marketing, but considering the abundance of misleading/poor informative videos on the internet, dentists and dental educators should be encouraged to share more accurate and educational content on social media platforms such as YouTube. On the other hand, it should not be forgotten that the use of social media in health-related fields may have some ethical and legal problems for both patients and dentists.

There are also some authors in the literature, who recommend having relevant healthcare professionals and government agencies to monitor social networks or using various validity-tools for video contents, in order to prevent misinformation on health-related issues to social media platforms on the internet. However, such an implementation has not been established so far [22,26].

There are problems such as limited /incorrect content and / or information that is not evidence-based in the videos reviewed about space maintainers on YouTube. For this reason, it is important for dentists to upload more and more evidence-based-videos and use an appropriate language that can be understood by patients/parents. Another important step is that those contents could be placed top on the "relevance" list, if possible, in order to attract the attention of the patients.

Like similar studies in the literature [7,10,27] which are examining YouTube content on health-related issues, the present study has several limitations. Firstly, YouTube is a highly dynamic platform, where videos are added and deleted on a daily basis. Therefore the results may change according to the date and time of the search. A longitudinal study may be more efficient in searching for YouTube content about space maintainers. Secondly, the results may vary according to the keywords used in the search. In order to evaluate the results more extensively, we used two keywords (space maintainer and space maintainers in pediatric dentistry) in the search. However, some patients might use other search terms and might achieve different results. The third and final limitation of the study is that the usefulness of videos is scored only by dentists. Different results might have been obtained if this scoring had been made by other healthcare professionals or patients/parents.

## Conclusions

YouTube contains various information on space maintainers. Information content increases as the video length increases. YouTube videos usually discuss etiology, purpose and general dental health effects of space maintainers. Namely, these videos are useful for parents to understand and visualize their function. But only a few videos mention their adverse effects, types and the importance of follow-up appointments. So, YouTube should not be used as a trusted source for the education of parents about space maintainers. Today, internet usage of patients is increasing. It is important for dental professionals and professional organizations to adapt to technology and upload videos with accurate and sufficient information to websites such as YouTube for improving the content of internet videos. In addition, dentists should give their patients sufficient information about the treatment process and warn their patients that videos on the internet may contain incorrect and limited information.

## Acknowledgments

I would like to thank Dr. Pelin Sezgin, Faculty of Dentistry at Biruni University, for her help in analyzing videos and critically reading the manuscript.

## Author Contributions

**Conceptualization:** Belen Şirinoğlu Çapan.

**Data curation:** Belen Şirinoğlu Çapan.

**Formal analysis:** Belen Şirinoğlu Çapan.

**Funding acquisition:** Belen Şirinoğlu Çapan.

**Investigation:** Belen Şirinoğlu Çapan.

**Methodology:** Belen Şirinoğlu Çapan.

**Project administration:** Belen Şirinoğlu Çapan.

**Resources:** Belen Şirinoğlu Çapan.

**Software:** Belen Şirinoğlu Çapan.

**Supervision:** Belen Şirinoğlu Çapan.

**Validation:** Belen Şirinoğlu Çapan.

**Visualization:** Belen Şirinoğlu Çapan.

**Writing – original draft:** Belen Şirinoğlu Çapan.

**Writing – review & editing:** Belen Şirinoğlu Çapan.

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
