## [Decision Letter · Decision Letter 0]

13 Jan 2021

PONE-D-20-32654

YOUTUBE AS A SOURCE OF INFORMATION ON SPACE MAINTAINERS FOR PARENTS AND PATIENTS

PLOS ONE

Thank you for submitting your manuscript to PLOS ONE. After careful consideration, we feel that it has merit but does not fully meet PLOS ONE’s publication criteria as it currently stands. Therefore, we invite you to submit a revised version of the manuscript that addresses the points raised during the review process.

We look forward to receiving your revised manuscript.

Kind regards,

Luigi Lavorgna

Academic Editor

PLOS ONE

Journal Requirements:

3.Thank you for stating the following financial disclosure:

 "No. The funders had no role in study design, data collection and analysis, decision to publish, or preparation of the manuscript."

4. We noticed you have some minor occurrence of overlapping text with the following previous publication(s), which needs to be addressed:

- https://onlinelibrary.wiley.com/doi/abs/10.1111/ipd.12277

- https://www.ajodo.org/article/S0889-5406(10)00488-9/fulltext

- https://onlinelibrary.wiley.com/doi/abs/10.1111/j.1365-263X.2008.00951.x

- https://onlinelibrary.wiley.com/doi/abs/10.1111/odi.12434

In your revision ensure you cite all your sources (including your own works), and quote or rephrase any duplicated text outside the methods section. Further consideration is dependent on these concerns being addressed.

Reviewers' comments:

Reviewer's Responses to Questions

**Comments to the Author**

1. Is the manuscript technically sound, and do the data support the conclusions?

Reviewer #1: Yes

Reviewer #2: Yes

2. Has the statistical analysis been performed appropriately and rigorously? 

Reviewer #1: Yes

Reviewer #2: Yes

3. Have the authors made all data underlying the findings in their manuscript fully available?

Reviewer #1: Yes

Reviewer #2: Yes

4. Is the manuscript presented in an intelligible fashion and written in standard English?

Reviewer #1: Yes

Reviewer #2: Yes

5. Review Comments to the Author

Reviewer #1: Çapan reported on YouTube videos on space maintainers, and evaluated the efficiency of videos by parents and patients. The manuscript is overall clear, and methods are sound. I only have some minor suggestions to the authors.

In the “Selection and Analyzes of Videos” section, the author mentions two authors reviewed the videos. However, in the current version of the manuscript, there is only one author. Please, double check whether all authors have been correctly added to the submission system.

It is not clear who actually scored the usefulness. Considering the objective of the study, it would be perhaps helpful multiple rating from healthcare professionals and parents. If this is not the case, authors should consider this as a limitation of the study.

I would recommend author adds a study flow diagram with number and reasons for exclusions for all YouTube videos.

In table 3 some rows look empty. Could you please double check and remove any useless rows?

In the discussion, authors should specifically comment on the active role healthcare providers should play on social media in order to provide complete and balanced information (e.g., Lavorgna et al. Mult Scler Relat Disord 2018; Lavorgna et al. Front Neurol 2020).

Reviewer #2: In this study çapan assessed the content of YouTube as an information source on space maintainers and evaluated the efficiency of videos by parents and patients. 

There are some minor revision needed, given in the specific comments below.

1. It is unclear in the paragraph "Usefulness and viewers' interaction" what the author means with "and 9 rated video".

2. If all very useful videos were uploaded by healthcare professionals or professional organizations, table 3 not is correct. It showed that 1 useful video was uploaded by Health information Web sites and not by healthcare professionals or professional organizations. Moreover, always in Table 3, the total numbers of videos do not correspond to the sum of each category. Maybe some rows are empty.

3. In light of the findings revealing that all very useful videos were uploaded by healthcare professionals or professional organizations, the author should discuss widely the relevance of the role of health care providers in uploading and supervising the health related information on online platforms.

6. PLOS authors have the option to publish the peer review history of their article (what does this mean?). If published, this will include your full peer review and any attached files.

Reviewer #1: No

Reviewer #2: No

---

## [Author Response · Author response to Decision Letter 0]

19 Jan 2021

Response to Reviewers

Reviewer #1: Çapan reported on YouTube videos on space maintainers, and evaluated the efficiency of videos by parents and patients. The manuscript is overall clear, and methods are sound. I only have some minor suggestions to the authors.

In the “Selection and Analyzes of Videos” section, the author mentions two authors reviewed the videos. However, in the current version of the manuscript, there is only one author. Please, double check whether all authors have been correctly added to the submission system.

• The videos were reviewed by two dentist. However, Dr. Sezgin only helped during the analyzing of the videos to ensure statistical reliability. As all the remaining planning, evaluation and writing of the study is my own, Dr. Sezgin was thanked at the acknowledements section. The expression "author", I used in the material-method section was changed to "dentist". 

It is not clear who actually scored the usefulness. Considering the objective of the study, it would be perhaps helpful multiple rating from healthcare professionals and parents. If this is not the case, authors should consider this as a limitation of the study.

• The usefulness of the videos was scored by dentists. Since the assesment of the study cannot be changed at the moment, in line with the suggestions of the reviewer, it was added to the limitations section that scoring with other healthcare professionals and parents may have different results.

I would recommend author adds a study flow diagram with number and reasons for exclusions for all YouTube videos.

• The number and reasons for exclusions for YouTube videos are given in Figure 1. 

In table 3 some rows look empty. Could you please double check and remove any useless rows?

• Table 3 is revised.

In the discussion, authors should specifically comment on the active role healthcare providers should play on social media in order to provide complete and balanced information (e.g., Lavorgna et al. Mult Scler Relat Disord 2018; Lavorgna et al. Front Neurol 2020).

• The discussion section has been revised in line with the suggestions of the reviewer.

Reviewer #2: In this study çapan assessed the content of YouTube as an information source on space maintainers and evaluated the efficiency of videos by parents and patients. 

There are some minor revision needed, given in the specific comments below.

1. It is unclear in the paragraph "Usefulness and viewers' interaction" what the author means with "and 9 rated video".

• A 9-rated video means a video which got full score. It is revised in manuscript to make the sentence clearer.

2. If all very useful videos were uploaded by healthcare professionals or professional organizations, table 3 not is correct. It showed that 1 useful video was uploaded by Health information Web sites and not by healthcare professionals or professional organizations. Moreover, always in Table 3, the total numbers of videos do not correspond to the sum of each category. Maybe some rows are empty.

• The expression error in results section and Table 3 is revised.

3. In light of the findings revealing that all very useful videos were uploaded by healthcare professionals or professional organizations, the author should discuss widely the relevance of the role of health care providers in uploading and supervising the health related information on online platforms.

• The discussion section has been revised in line with the suggestions of the reviewer.

---

## [Editor Report · Decision Letter 1]

20 Jan 2021

YOUTUBE AS A SOURCE OF INFORMATION ON SPACE MAINTAINERS FOR PARENTS AND PATIENTS

PONE-D-20-32654R1

We’re pleased to inform you that your manuscript has been judged scientifically suitable for publication and will be formally accepted for publication once it meets all outstanding technical requirements.

Kind regards,

Luigi Lavorgna

Academic Editor

PLOS ONE
---

## [Editor Report · Acceptance letter]

29 Jan 2021

PONE-D-20-32654R1 

YouTube as a source of information on space maintainers for parents and patients 

Dear Dr. Çapan:

I'm pleased to inform you that your manuscript has been deemed suitable for publication in PLOS ONE. Congratulations! Your manuscript is now with our production department. 

Kind regards, 

on behalf of

Dr. Luigi Lavorgna 

Academic Editor

PLOS ONE